# Take-Home Messages from the COVID-19 Pandemic: Strengths and Pitfalls of the Italian National Health Service from a Medico-Legal Point of View

**DOI:** 10.3390/healthcare9010017

**Published:** 2020-12-25

**Authors:** Matteo Bolcato, Marco Trabucco Aurilio, Anna Aprile, Giulio Di Mizio, Bruno Della Pietra, Alessandro Feola

**Affiliations:** 1Legal Medicine, Department of Molecular Medicine, University of Padua, 35121 Padova, Italy; anna.aprile@unipd.it; 2Department of Medicine and Health Sciences “V. Tiberio”, University of Molise, 86100 Campobasso, Italy; marco.trabuccoaurilio@unimol.it; 3Forensic Medicine, Department of Law, “Magna Graecia” University of Catanzaro, 88100 Catanzaro, Italy; giulio.dimizio@unicz.it; 4Department Experimental Medicine, University of Campania “Luigi Vanvitelli”, 80138 Naples, Italy; bruno.dellapietra@unicampania.it (B.D.P.); alessandro.feola@unicampania.it (A.F.)

**Keywords:** COVID-19, medical liability, ethics, medico-legal evaluation, patient blood management, clinical risk management

## Abstract

The World Health Organization (WHO) declared the outbreak of the Coronavirus disease-2019 (COVID-19) infection a pandemic on 11 March 2020. As of the end of October 2020, there were 50 million cases of infection and over one million deaths recorded worldwide, over 45,000 of which occurred in Italy. In Italy, the demand for intensive care over the course of this pandemic crisis has been exceptionally high, resulting in a severe imbalance between the demand for and availability of the necessary resources. This paper focuses on elements of preventive medicine and medical treatments in emergency and non-emergency situations which, based on the international scientific literature, may prove to be useful to physicians on a behavioral level and avert professional liability problems. In order to achieve this objective, we have performed a search on MEDLINE to find published articles related to the risks associated with the pandemic that contain useful suggestions and strategies for mitigating risks and protecting the safety of the population. The results have been collocated in line with these specific study areas.

## 1. Introduction

The World Health Organization (WHO) declared the outbreak of the Coronavirus disease-2019 (COVID-19) infection a pandemic on 11 March 2020. The infection can cause severe respiratory insufficiency, sometimes requiring hospitalization in an intensive care unit (ICU) in order to provide invasive ventilatory support [1]. The outcome may be fatal, despite the use of considerable professional and economic resources, equipment, and facilities, especially in older patients and with the development of acute respiratory distress syndrome (ARDS). In one particular cohort of 1591 patients admitted to ICU in Lombardy between 20 February and 18 March 2020, the mortality rate was 26% [2]. The pandemic has attacked and, in many cases, deeply unsettled various aspects of our view of and relationship with life, society, and rights, raising critical issues and limitations that were previously believed to have been totally overcome. Nevertheless, Italy recognizes and guarantees every individual the protection of fundamental rights, including the right to health (Article 32 of the Constitution), the right to safety (Article 16), and the right to personal freedom (Article 13). In addition, the Italian legal system contains various provisions detailing how to avail oneself of these rights, including the right to refuse medical assistance, even if the purpose is to prolong life [3,4,5,6,7].

Despite efforts on hospital and national levels to raise the quality and efficacy of healthcare, the demand for intensive care over the course of this pandemic crisis has been exceptionally high, resulting in a severe imbalance between the demand for and availability of the necessary resources. This situation has raised unprecedented dilemmas and situations [8], which the Italian Society of Anesthesia, Analgesia, Resuscitation and Intensive Care (SIAARTI) addressed in a document entitled: “Clinical ethics recommendations for the allocation and suspension of intensive care treatments in exceptional, resource-limited circumstances” [9]. This document provides a concrete analysis of the issue which involves a complex and detailed procedure. It also proposes a method for the ethical-scientific evaluation of ICU admission criteria, described by the various healthcare professionals and operators involved [10].

Furthermore, to stem the spread and severity of the pandemic, national and European authorities issued numerous emergency provisions of a logistical and economic nature to suspend production activities and movement, including a general lockdown on economic and social activities that lasted until the beginning of May 2020 [11]. Thereafter began a new phase of coexistence with the virus in circulation with protection measures in place mostly consisting of personal hygiene, protection of airways using masks, and social distancing.

As of the end of October 2020, there were 50 million cases of infection and over one million deaths recorded worldwide, over 45,000 of which occurred in Italy [12,13].

The epidemiological situation in Italy seems to be relatively under control, despite the slight and continual increase in cases of infection.

New and significant challenges have emerged due to the need to coexist with the virus until a sufficient population immunity threshold has been reached, primarily through vaccination. This paper focuses on elements of preventive medicine and medical treatments in emergency and non-emergency situations which, based on the international scientific literature, may prove to be useful to physicians on a behavioral level and avert professional liability problems. Furthermore, as part of this literature review, we have analyzed articles that contain recommendations of strategies and conduct to mitigate pandemic-related risks and to protect the population.

## 2. Materials and Methods

### Search Strategy

We searched for all articles written in English in the literature available on MEDLINE in the last year until 31 October 2020 using the following search terms: “medical liability SARS-CoV-2,” “medical liability COVID-19,” “medical liability pandemic,” “malpractice SARS-CoV-2,” “malpractice COVID-19,” “medico-legal litigation SARS-CoV-2,” “medico-legal litigation COVID-19,” “liability claims SARS-CoV-2,” and “liability claims COVID-19.” The objective of this review is to concentrate on articles and documents that contain useful information on prevention techniques for containing dissemination and contamination within the broader program of clinical risk management. We included only articles available in the literature that contain information concerning pandemic-related situations of risk and that recommend the implementation of specific activities designed to mitigate these risks and facilitate the proper operation of strategic sectors to protect the population. Articles that contained merely clinical or microbiological indications regarding the nature and characteristics of the virus were excluded. All sources were evaluated independently by two of the authors to determine their relevance to the present study and then selected for inclusion by all the authors.

## 3. Results

Using the above search strings, MEDLINE produced 53 results that the authors considered relevant to the study in that they contain useful information for the prevention of adverse events despite the persistence of the pandemic. Details of the documents analyzed and excluded are displayed in PRISM format in Figure 1. In order to facilitate comprehension of the text, we have collocated the results according to the specific study areas and reviewed them in a paragraphical narrative.

### 3.1. Healthcare and Safety of Care

In the course of a pandemic, reaching a high level of care safety [14,15] may prove to be a complex challenge [16]. It is clear that healthcare professionals have concerns regarding professional liability disputes that arise due to pandemic-related issues [17,18,19]. During this particular pandemic, several countries have contemplated creating a legal shield that limits the professional liability of healthcare personnel throughout this emergency situation [20,21]. More specifically, in March 2020, pending the approval of the Cura Italia Decree, certain amendments were issued stipulating that a healthcare worker shall be held liable only if the alleged harm caused to the patient is caused directly by malicious misconduct or gross negligence, where gross negligence represents an unjustified violation of the basic principles that govern the healthcare profession or of any emergency protocols established to handle the situation [22,23,24,25,26]. Furthermore, it was feared that various profiles of liability may ensue in connection with the following scenarios:(a)Lack of or reduced ability to care for and treat infected patients due to the insufficiency of available resources, which has contributed to causing harm or patient death.(b)Harm to or death of patients not infected by COVID-19 but affected by ingravescent diseases (e.g., tumors or cardiopathy) indirectly caused by the pandemic, the reduction of available healthcare resources, or the cessation of medical activities pertaining to non-pandemic-related infirmities preventing access to adequate, timely care.(c)Patient infection within hospitals or residential facilities used as shelters due to failure to observe prevention recommendations and protocols.(d)Infection and consequent harm to or death of employees in ill-prepared companies, or more likely, hospitals and residential care homes that lack adequate safety equipment.(e)Harm to or death of patients infected with COVID-19 and treated with experimental or off-label drugs.

Before analyzing each of the hypothetical profiles listed above, it is worth remembering that from a general point of view the Italian legal system, even without specific liability exemption laws, contains certain important institutions that relate to this situation. Regardless of the specific liability profile envisaged, the fact that there is no scientific evidence available regarding the nature, transmissibility, and treatment for the SARS-CoV-2 infection necessitates careful consideration. At the time this document was drafted and even less so towards the beginning of the epidemic, there were no guidelines or recommendation documents based on solid evidence that could provide reasoned and specific direction to guide medical decisions. In effect, with no covering laws, this meant extreme difficulty in delineating either proper or censurable conduct with scientific certainty.

The second aspect of interest concerns the effect a substantial lack of hospital resources has on the decisions made by individual operators, liability for which in no way attributable to the operator personally but to institutional decision-makers. This is a particularly significant factor in criminal proceedings where liability is necessarily attributed to a specific person.

The profile of liability described in point (a), where an individual operator is unable to administer life-saving treatment due to insufficient resources [27,28], represents the legal scenario of inability to perform a service for reasons not attributable to the healthcare professional. The professional would therefore be exempted from any compensation obligation pursuant to Articles 1218 and 1256 of the civil code. These laws are more easily applied within the context of personal liability as opposed to the liability of facilities, which are obliged to demonstrate their inability to fulfill their duties due to (objectively) unpredictable and unavoidable causes [29,30], to a crisis and to an inability to ensure the standard of care previously delivered [31,32,33,34,35]. To that end, Article 2236 of the civil code may be invoked, which sanctions that “*If performing the service requires the solution to particularly difficult technical problems, the service provider is not liable for damages in the absence of malicious misconduct or gross negligence.*” According to some authors, this provision could serve as an initial shield against possible attacks, especially if brought for speculative purposes. Furthermore, the provision set forth in Article 9 of Law No. 24/2017 on recourse states: “*In order to quantify damages, with the exception of the provisions set forth in Law No. 20, 14 January 1994, Art. 1 (1b) and Art. 52 (2) of the consolidated act referred to in Royal Decree No. 1214, 12 July 1934, particularly difficult situations, including those of an organizational nature, that affect public health or social care facilities in which healthcare professionals operate must be taken into account (…).*” Although said provision refers to recourse claims, it is clear that it also implies that certain organizational difficulties may enable the quantum of compensation to be regulated. The epidemic that has developed in Italy can without doubt be classified as a medical, scientific, and organizational situation of exceptional difficulty.

In light of these provisions, although an ad hoc law to limit professional medical liability may not seem necessary, high-level legal assessment and medico-legal abilities [36], compliant with the methodological and scientific precision required [37,38], are essential.

### 3.2. Future of the Healthcare System

However, there is still the issue of liability for the decisions of those who have caused the national situation of general unpreparedness and the constant reduction of social and public healthcare services by failing to implement policies that foresee both developing and unanticipated exigencies and situations. Such shortsightedness, the reasons and associated liability for which shall be analyzed, has not only led to a failure to safeguard human dignity, given in many cases it is impossible to allocate adequate treatment resources to some patients, but also, no less dramatically, to healthcare professionals being obliged to implement decisions they would never have wanted to implement.

Though this situation has defined the initial stages of the pandemic, it cannot be allowed to become the norm in future crisis stages. For that reason, regional and national initiatives must be activated to increase available therapeutic resources in preparation for a further rise in demand. Such an increase may require action on three levels: (1) intensive care, (2) internal medicine, and (3) community care.

As regards the first level, national institutions have made significant contributions in the past few months. Prior to the pandemic (2017), there were 5090 intensive care spaces available in Italy (8.42 per 100,000 inhabitants) [39], whereas the Ministry of Health is currently aiming to reach 8500 spaces by October. This increase may serve to prevent situations where a number of patients have no access to intensive care and the potential medical, social, and legal repercussions. These statistics are calculated on a national basis, but, as was noted in the first wave in March and April 2020, the pandemic did not affect all areas of the country with equal intensity [40]. Although the Italian National Health Service is organized on a regional basis, active and effective cooperation between all regions is essential in order to put available resources to the most effective use on a national level.

Treatment during the pandemic has required not only intensive care unit admission but also considerable assistance from entire medical and surgical departments, which have been transformed into reception areas for infected patients. This influx of patients has had two basic effects: (1) interruption of surgical care and treatment of other diseases that do not require urgent intervention, and (2) filling departments with infected patients [41,42,43,44]. This has affected certain geographical areas more than others and involved assisting and treating many patients. However, in the event the pandemic were to place a further substantial burden on the health service, employing the same measures as were adopted a few months ago is not an option in that it is not feasible, due to public health and ethical-legal liability concerns as regards sick patients, especially the chronically ill, to impose another substantial suspension on ordinary activities [45,46,47].

With respect to point (b), in order to prevent this legal scenario, home medical care services for patients infected with COVID-19 who do not need hospital or intensive care will have to be enhanced. This approach has been incentivized through the extensive hiring of healthcare personnel for community care, especially nurses, who can provide support to sick people and their families in isolation at their homes or in designated facilities to prevent the spread of infection and enable full physical recovery. It will also prevent hospital congestion and facilitate the normal practice of non-pandemic-related medical and surgical services [48]. This level of organization requires action on two fronts: (1) application of strict protocols to ensure patient safety, and (2) implementation of systemic strategies for handling acute shortages.

As regards point (1), accurate assessments of patient conditions are essential so as to decide whether to hospitalize or whether home care would be sufficient. This assessment should be carried out by trained medical personnel and subjected to periodic reviews by means of remote monitoring systems so that if the situation were to deteriorate, the patient could be hospitalized in time and, if necessary, admitted to intensive care [49,50]. Therefore, shared operating protocols are needed both to assist practitioners and to integrate models of conduct to prevent professional and organizational liability implications.

### 3.3. Patient Blood Management

In relation to point (2), plans and strategies to manage severe crisis situations must be formulated on an institutional level, even when resources are available, making full use of all the options that scientific innovation presents. As a result, in times of greater difficulty, all available technical solutions can be utilized to ensure the continuity of essential medical-surgical activities and to develop not only logistical and economic but also scientific responses to crises. An example of the type of proactivity needed to ensure such continuity during emergency situations would be the proper management of blood resources. Maintaining adequate supplies of safe blood in order to guarantee the execution of necessary surgical procedures or other medical activities can prove to be particularly difficult in emergency situations, especially epidemics. In the first few months of the pandemic, many countries [51], including Italy [52], experienced a consistent decrease in donations and, therefore, in the availability of hematic products. Although to date there is no evidence to suggest the possibility of virus transmission through blood from symptomatic individuals, the shortage is due to the greater difficulty in travel, the risks associated with congregating in donation centers, and a higher percentage of the population that is potentially infected and unable to donate. This decrease has not produced devastating consequences on the population and on medical activities, since the decision to postpone non-urgent surgeries was taken simultaneously and, therefore, the demand for hematic products decreased considerably [53,54,55]. Nevertheless, this new-found balance cannot be maintained; routine activities cannot be interrupted again at the expense of public health. A systematic, proactive, and organized approach is needed to meet this type of demand such as patient blood management (PBM).

PBM is a multidisciplinary, multimodal approach adopted to limit or eliminate the need for allogenic blood transfusions through evidence-based management of anemia, blood loss reduction, and optimization of blood salvage strategies [56]. This approach comprises over 100 methods and activities with the aim of optimizing the use of the patient’s blood instead of resorting to allogenic blood transfusions [57,58]. In addition to reducing transfusion risks [59], PBM is also proving instrumental in applying the law on patient safety in Italy [60,61]. It does, however, necessitate consistent and systemic planning and management if it is to be a dependable resource in times of severe crisis [62,63,64,65].

The implementation of PBM presents both a challenge and a particularly profitable opportunity in Italy and other countries with an older average population. In the years ahead, these countries will find themselves unable to satisfy the transfusion demand, as the population will not be in a position to donate blood due to old age despite their potential need to receive it. In that sense, the pandemic may accelerate the instituting of processes to safeguard public health in addition to presenting a remarkable opportunity for economic saving [66].

### 3.4. Residential Care Homes

With regard to point (c), in recent months there has been significant media attention, particularly because of the high number of deaths in residential care homes for the elderly due to, according to critics, superficial management of admissions and preventative activities to create a safe environment for the frail that occupy it [67]. It is important to state that the prevention of respiratory infections in such contexts is particularly complex and poses a challenge [68,69] on several fronts: firstly, the frailty of elderly patients in conjunction with the scarcity of clinical manifestations which can present late and are thus difficult to detect in a timely manner. Secondly, nursing homes often lack the specialized personnel and equipment needed to deal with acute events [70]. In emergency situations, many residential facilities in Italy have had to treat and assist infected patients who were not promptly hospitalized. This can easily create exposure to medico-legal litigation. Therefore, it is imperative that risk management programs be set up in every facility in order to deal with this pandemic effectively [71].

While the level of unpreparedness during the initial stages of the pandemic may have been justifiable, the current situation necessitates that there be no further procrastination in the implementation of strategies to improve care by means of early diagnosis of signs and symptoms, the use of standard diagnostic criteria, the adoption of adequate preventive measures, and the institution of monitoring and control systems. Such measures will simultaneously increase the probability of optimizing results and ensure the safety of patients and the well-being of workers [72,73]. Further indications include the institution of internal coordination committees, periodic employee and patient screening programs, careful selection of personal protective equipment (PPE), promotion of training initiatives that all members of staff must attend as well as the necessary logistical activities for organizing isolation and monitoring procedures.

From a medico-legal standpoint, each facility needs to prepare internal protocol documents with clear behavioral indications for all operators, including a precise plan of action to guarantee safe crisis management. The objective of such documents, registered and shared with all facility personnel, is to ensure that all those concerned act responsibly and to provide an effective, documented defense in the event of litigation [74,75,76,77].

### 3.5. Safety of Employees

Scenario (d) highlights the need to ensure the safety of employees, especially healthcare employees, as they are deemed at greater risk of infection. On 24 March 2020, an agreement between the Italian government and representatives for companies and employees was proposed and subsequently adopted on 24 April 2020 [78]. The main recommendations of the agreement have been summarized in Table 1.

The Italian government supports the implementation of these recommendations as they promote the safety and protection of workers, forming the basis for the safety activities that companies must adopt, including for the purposes of preventing litigation. Healthcare personnel must be included among the category of workers who, being in constant contact with infected patients, are at greater risk of infection. This factor was particularly highlighted during the initial weeks of the epidemic due to the severe shortage of PPE. This brings to mind the high number [79] of physicians who died of COVID-19 infection and the National Institute for Insurance against Accidents at Work (INAIL) declaration in favor of compensating those affected [80].

Finally, as far as liability in the event of harm or death caused by the use of experimental or off-label drugs is concerned, Italian law permits and delimits their usage (Law No. 94/98). Despite the fact that the use of off-label drugs may expose a patient to potential risks, in the specific context of the pandemic—considering the scarce knowledge of the virus’ mechanism of action and target organs—physicians must show greater prudence in resorting to such drugs. Evaluations of proper conduct shall take into account the specifics of the case, the severity of the patient’s clinical manifestations, the nature of the drug, known contraindications, verified side effects, and the dosage administered.

### 3.6. Administration of Justice and Scientific Research

Since the beginning of the year, Italy has experienced a lockdown that is even more dangerous from a social viewpoint—that of the justice system. Particularly, the justice system has been paralyzed and come to a standstill with the exception of cases of extreme urgency and gravity. This has caused a series of problems with regard to maintaining the country’s social and economic system. The motive for the lockdown is the fear of potential infection in courtrooms and of court employees. However, as things stand, the justice system must be reactivated as soon as possible in order to prevent further delays in the country’s economic activities. To do so safely, courts can take advantage of all the available technological and organizational resources at their disposal, e.g., videoconferencing systems, certified electronic signatures, and remote access to judicial archives [81]. In so doing, the justice system will see that the majority of the activities in its remit can be performed using IT systems.

Equally important, to provide an adequate response to major emergencies, particularly to epidemics of mostly unknown infectious agents, it is essential to seek the input of all scientific disciplines, especially that of legal medicine. In this regard, it is useful to reiterate the value of autopsies from a legal and, more importantly, scientific and clinical point of view in identifying causes of death and in contributing to the reconstruction of physiopathological pathways that lead to death. This approach may provide determining scientific information to find possible therapeutic options that would otherwise have been impossible to obtain [82,83,84,85]. In order to do that, methodology and adherence to strict protocols in all analyses and activities are required [86].

Ultimately, Italy must adopt a proactive mentality towards these and other crises if it is to be prepared and scientifically equipped to handle future major emergencies.

## 4. Conclusions

The COVID-19 pandemic has highlighted several pitfalls in the National Health Service in Italy and, more generally, in Western countries that have had to handle and find extemporaneous solutions to this unforeseen event (e.g., temporary recruitment of physicians and nurses, construction of new departments and even new hospitals, adaptation of existing facilities, and procurement of PPE). As a result, the health services have been faced with ethical dilemmas such as deciding who to treat considering the shortage of resources and inability to ensure all patients receive the necessary intensive care treatment [87,88,89,90,91,92], dilemmas which Western–particularly the Italian–health services did not envisage having to face again. In that sense, having a clear understanding of the ethical principles that govern medical actions means distributive justice, nonmaleficence, respect for patient autonomy and dignity regardless of their degree of vulnerability, and confidentiality of medical data in order to make the most appropriate decisions for patients [87,88].

Initially, public opinion was in favor of instituting a legal shield to protect physicians, but the provision encountered numerous obstacles, abroad as well as in Italy, and to date has not been approved. Clarification is necessary since it is proper to safeguard healthcare professionals who have been forced to serve in dramatic conditions (shortage of PPE, lack of knowledge regarding the etiopathogenesis of the infection, lack of tried-and-tested treatments) but also to recognize organizational and medical flaws that have de facto exacerbated the situation.

The current pandemic has led to profound changes in the lives of individuals and healthcare professionals, brought to light weaknesses in the more developed health services, and called into question many healthcare policy decisions that have been made in recent decades. There are obvious limits to this study as many articles have been published in recent months and, therefore, offering a complete and up-to-date review of the literature presents a challenge. Although not all areas of pandemic-related risks can be mentioned, this article has analyzed and explored strategies for mitigating risks associated with the dissemination of the infection in various key sectors in the administration of public services for immediate application. In addition, it has highlighted programs that may aid the National Health Service such as PBM, particularly important in Western countries with an aged population. Some countries have already implemented praiseworthy organizational strategies: for example, Australia has put in place an extensive PBM program, such that has facilitated a 60% reduction in the use of red blood cells [66,93]. These results show that from a medical point of view, some countries are more prepared than Italy to handle a major emergency. It must be said, however, that other European countries have encountered similar problems to Italy [94].

In addition to all the issues discussed in this article, the future will present other issues in connection with the development of a vaccine. It seems that at least one vaccine will be available from January 2021 and nationwide distribution will soon commence. This may create further problems in that the probability of procuring sufficient doses for the entire population is remote, meaning difficult decisions will need to be made in terms of who to prioritize. Moreover, additional vaccines may become available in the coming months, resulting in a diversified vaccination campaign due to the particular characteristics of each product in terms of efficacy and protection time. The health authorities will need to weigh these elements carefully, providing full decisional transparency. The public should also be apprised of these decisions by means of an information campaign in order to maximize voluntary participation in the vaccination drive.

## Figures and Tables

**Figure 1 healthcare-09-00017-f001:**
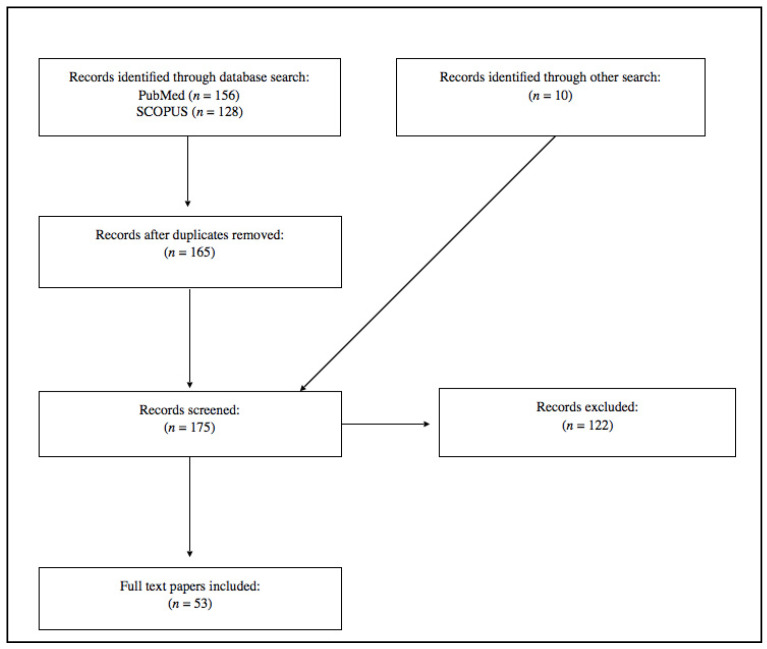
Flowchart depicting the choice of studies.

**Table 1 healthcare-09-00017-t001:** Main recommendations listed in the agreement between the Italian government and representatives for companies and employees signed on the 24 March 2020.

**Information**	Using the most appropriate and effective methods, the company shall inform all employees, and any who enter the premises, of the provisions to contain the spread of the virus and of the rules to be followed. This information should include: the obligation to stay at home in the event of a fever above 37.5 °C, maintain safe distances, observe handwashing and hygiene rules, and to inform one’s employer of any flu-like symptoms that appear during the work period in a timely and responsible manner.
**Methods for accessing the workplace and checks that shall be performed**	Measurement of body temperature, triage.
**Hygiene in the workplace**	The company shall ensure the daily cleaning and periodic sanitization of workstations, common rooms/areas, and entertainment facilities. The company shall provide appropriate handwashing detergents and recommend frequent handwashing with soap and water. In the event a certain work-related activity necessitates proximity between workers of less than one meter and no other organizational solutions are available, the use of masks and other protective equipment is mandatory.
**Handling a symptomatic person in the workplace**	In the event someone present in the workplace develops a fever and symptoms of respiratory infection such as a cough, said person must report to the personnel office immediately and be isolated. The company shall immediately inform the competent healthcare authorities and call the COVID-19 emergency numbers.
**Role of the occupational health physician**	The doctor shall assist the company in prevention activities and screening, if required, of employees as well as reporting potentially fragile employees.

## Data Availability

The datasets used during the current study are available from the corresponding author upon reasonable request.

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
