# Peer review of "Take-Home Messages from the COVID-19 Pandemic: Strengths and Pitfalls of the Italian National Health Service from a Medico-Legal Point of View"

_healthcare, 2020, doi:10.3390/healthcare9010017_

Round 1
Reviewer 1 Report
This is an important article researching elements that might help emergency and non-emergency clinicians reduce the medico-legal liability in these unprecedented pandemic situation. It also assessed risk mitigation messages and strategies which is increasingly important as the world struggles with a resurgence of the virus in many locations. Although the article is only relevant to the Italian Health service, it is still an important piece of work that merits publication.
Author Response
we want to thank the reviewer for his work and comments on our article
Reviewer 2 Report
This review article by Bolcato et al. focused on the COVID-19 pandemic and its pitfalls of the Italian National Health Service. Authors reviewed it from a medico-legal point of view. Since COVID-19 is currently a topic worldwide and there are several problems as authors mentioned, the concept of this review article manuscript is valuable. Although this article seems written well, authors may want to consider several issues as follows.
Major comments;
1) When authors search for articles, the term SARS-CoV-2 should be included because COVID-19 is caused by Acute Respiratory Syndrome Coronavirus-2 (SARS-CoV-2) infection. Otherwise, articles focused on COVID-19 are not all searched.
Minor comments;
1) Authors should describe full terms of COVID-19 at the first time of use as Coronavirus disease-2019 (COVID-19).
Reviewer 3 Report
This paper offers great contributions to the field healthcare. I think that you should consider including some possible limitations to your paper.
This article does a great job at explaining the many challenges to the Italian healthcare system associated with the COVID-19 pandemic. I think that the authors should consider adding a discussion of the situation in other similar Western nations, just as a comparison to the Italian situation. It would be interesting to see how other developed nations' strategies compare the Italian response. The COVID-19 vaccine is scheduled to arrive in Italy in January. I think that a discussion about what the vaccine will mean for the COVID-19 situation in Italy would be an interesting addition to this paper. Also, the authors don't mention any limitations to their work. This is important in any research paper.Author Response
please see the attachment

Round 2
Reviewer 2 Report
This review article by Bolcato et al. focused on the COVID-19 pandemic and its pitfalls of the Italian National Health Service. Authors reviewed the manuscript appropriately according to the review’s suggestions. It appeared better. I have no further comment.